

# Achilles and tail tendons of perlecan exon 3 null heparan sulphate deficient mice display surprising improvement in tendon tensile properties and altered collagen fibril organisation compared to C57BL/6 wild type mice

Cindy C. Shu[1], Margaret M. Smith[1], Richard C. Appleyard[2,3], Christopher B. Little[1,4] and James Melrose[1,4,5]

[1] Raymond Purves Bone and Joint Laboratory, Kolling Institute of Medical Research, University of Sydney, Australia

[2] Murray Maxwell Biomechanics Laboratory, Royal North Shore Hospital, University of Sydney, St. Leonards, New South Wales, Australia

[3] Surgical Skills Laboratory, Australian School of Advanced Medicine, Macquarie University, Sydney, New South Wales, Australia

[4] Sydney Medical School, Northern, University of Sydney, Sydney, Australia

[5] Graduate School of Biomedical Engineering, University of New South Wales, Sydney, New South Wales, Australia

Corresponding author
James Melrose,
james.melrose@sydney.edu.au

## ABSTRACT

The aim of this study was to determine the role of the perlecan (Hspg2) heparan sulphate (HS) side chains on cell and matrix homeostasis in tail and Achilles tendons in 3 and 12 week old $Hspg2$ exon 3 null HS deficient ($Hspg2^{\Delta3-/\Delta3-}$) and C57 BL/6 Wild Type (WT) mice. Perlecan has important cell regulatory and matrix organizational properties through HS mediated interactions with a range of growth factors and morphogens and with structural extracellular matrix glycoproteins which define tissue function and allow the resident cells to regulate tissue homeostasis. It was expected that ablation of the HS chains on perlecan would severely disrupt normal tendon organization and functional properties and it was envisaged that this study would better define the role of HS in normal tendon function and in tendon repair processes. Tail and Achilles tendons from each genotype were biomechanically tested (ultimate tensile stress (UTS), tensile modulus (TM)) and glycosaminoglycan (GAG) and collagen (hydroxyproline) compositional analyses were undertaken. Tenocytes were isolated from tail tendons from each mouse genotype and grown in monolayer culture. These cultures were undertaken in the presence of FGF-2 to assess the cell signaling properties of each genotype. Total RNA was isolated from 3–12 week old tail and Achilles tendons and qRT-PCR was undertaken to assess the expression of the following genes $Vcan$, $Bgn$, $Dcn$, $Lum$, $Hspg2$, $Ltbp1$, $Ltbp2$, $Eln$ and $Fbn1$. Type VI collagen and perlecan were immunolocalised in tail tendon and collagen fibrils were imaged using transmission electron microscopy (TEM). FGF-2 stimulated tenocyte monolayers displayed elevated $Adamts4$, $Mmp2$, $3$, $13$ $mRNA$ $levels$ compared to WT mice. Non-stimulated tendon $Col1A1$, $Vcan$, $Bgn$, $Dcn$, $Lum$, $Hspg2$, $Ltbp1$, $Ltbp2$, $Eln$ and $Fbn1$ $mRNA$ $levels$ showed no major differences between the two genotypes other than a decline with ageing

while LTBP2 expression increased. Eln expression also declined to a greater extent in the perlecan exon 3 null mice ($P < 0.05$). Type VI collagen and perlecan were immunolocalised in tail tendon and collagen fibrils imaged using transmission electron microscopy (TEM). This indicated a more compact form of collagen localization in the perlecan exon 3 null mice. Collagen fibrils were also smaller by TEM, which may facilitate a more condensed fibril packing accounting for the superior UTS displayed by the perlecan exon 3 null mice. The amplified catabolic phenotype of $Hspg2^{\Delta3-/\Delta3-}$ mice may account for the age-dependent decline in GAG observed in tail tendon over 3 to 12 weeks. After Achilles tenotomy $Hspg2^{\Delta3-/\Delta3-}$ and WT mice had similar rates of recovery of UTS and TM over 12 weeks post operatively indicating that a deficiency of HS was not detrimental to tendon repair.

## INTRODUCTION

Heparan sulphate (HS) is an ancient glycosaminoglycan (GAG) which has evolved over hundreds of millions of years of vertebrate and invertebrate evolution (*Yamada, Sugahara & Ozbek, 2011*). HS has developed important cell regulatory and interactive properties with matrix components which stabilize the extracellular matrix (ECM) and maintain tissue homeostasis (*Whitelock & Iozzo, 2005*). HS is attached to a number of matrix and cell associated proteoglycans (PGs) including, perlecan, agrin, type XVIII collagen, syndecan and glypican (*Gallagher, 2015*; *Iozzo & Schaefer, 2015*). Perlecan is an important matrix organizational, stabilizing and cell-signaling hub in tissues. Besides its biodiverse range of interactive ECM components perlecan-HS also binds and delivers a number of growth factors such as fibroblast growth factor (FGF)-2, 7, 9, 18; platelet derived growth factor (PDGF); *Wnt* (a condensation of terms describing the *Winged* and *Int* transcription factor morphogens); Sonic Hedgehog (*SHH*); vascular endothelial growth factor (VEGF),and bone morphogenetic proteins (BMPs) to their cognate receptors (*Whitelock, Melrose & Iozzo, 2008*). The aim of the present study was to ablate these HS chains by deletion of exon 3 of perlecan core protein and determine what effect this had on the homeostasis and function of tendon. A level of redundancy is normally evident in physiological systems thus we envisaged that we should also see what other molecules assisted the HS chains of perlecan in such processes which maintain tissue functionality and homeostasis. As already noted, HS also occurs on a number of proteoglycans other than perlecan however it is not known to what extent these can fill-in for a deficit in the perlecan-HS chains. Our experimental design also allowed us to ascertain what accessory roles these may have in the maintenance of tissue function and homeostasis.

Tendons are tough semi-elastic force transmitting cable-like structures between bone and muscle (*Benjamin, Kaiser & Milz, 2008*; *Bey & Derwin, 2012*; *Wang, Guo & Li, 2012*). Type I collagen is the major tendon fibrillar collagen providing it with mechanical strength (*Screen et al., 2015*). Elastin is a minor tendon component (2–4% w/w dry weight) localized
around tenocytes and between collagen fascicles (*Grant et al., 2013*). Elastin associated microfibrillar proteins form an integrated microstructural network along with perlecan and type VI collagen (*Jensen, Robertson & Handford, 2012*) forming a mechanosensory system whereby tenocytes perceive and respond to perturbations in their mechanical micro-environments to achieve tissue homeostasis (*Pang et al., 2017*). Type V and XI collagen are minor internal components of I/V/XI heterofibrils (*Linsenmayer et al., 1993*; *Nurminskaya & Birk, 1998*; *Smith, Zhang & Birk, 2014*; *Wenstrup et al., 2004*; *Wenstrup et al., 2011*). HS binding sites in type V and XI collagen have roles in tendon nucleation and lateral stabilization of collagen fibrils (*LeBaron et al., 1989*; *Ricard-Blum et al., 2006*). Type VI and XI collagen are associated with the cell surface of intervertebral disc cells (*Hayes et al., 2016*) and articular chondrocytes (*Carvalho et al., 2006*; *Horikawa et al., 2004*; *Wilusz, Sanchez-Adams & Guilak, 2014*; *Zelenski et al., 2015a*; *Zelenski et al., 2015b*). Localisation of type XI collagen at the cell surface is HS dependant (*Petit et al., 1993*; *Smith Jr, Hasty & Brandt, 1989*). Perlecan and elastin also colocalise at the cell surface of disc cells, tenocytes and chondrocytes (*Hayes et al., 2011a*; *Hayes et al., 2011b*; *Yu & Urban, 2010*) and have roles in the assembly of elastic microfibrils at the cell surface which have mechanosensory and mechanotransductive functions which direct tissue homeostasis (*Grant et al., 2013*; *Hayes et al., 2016*). The small leucine repeat proteoglycan (SLRP) family also interact with collagen fibres (*Schonherr et al., 1995*; *Svensson, Narlid & Oldberg, 2000*; *Viola et al., 2007*). The leucine rich repeat (LRR) domains of SLRPs wrap around the collagen fibre exposing their GAG side chains which interact with the SLRP-GAG chains on adjacent collagen chains providing lateral stabilization (*Islam et al., 2013*; *Kalamajski, Aspberg & Oldberg, 2007*; *Svensson, Heinegard & Oldberg, 1995*). Tenocytes express the HS-proteoglycan, perlecan (*Hspg2*) which colocalizes with type VI collagen (*Hayes et al., 2016*; *Wilusz, Defrate & Guilak, 2012*; *Wilusz, Sanchez-Adams & Guilak, 2014*; *Zelenski et al., 2015b*). Atomic force microscopy (AFM) studies have identified biomechanical roles for these pericellular components with perlecan providing a level of compliancy which may be cytoprotective (*Li et al., 2015*; *McLeod, Wilusz & Guilak, 2013*; *McNulty & Guilak, 2015*; *Plodinec, Loparic & Aebi, 2010*; *Sanchez-Adams, Wilusz & Guilak, 2013*; *Taffetani et al., 2015*; *Wang et al., 2012*). Perlecan is a minor proteoglycan in normal tendon but when tendon is damaged such as in a rotator cuff tendinosis model (*Melrose et al., 2013*) the tenocytes dramatically increase their production of perlecan suggesting that it participates in tissue repair. In the present study we were interested in ascertaining how ablation of the HS chains in perlecan of *Hspg2* exon 3 null mice affected tendon organization and functional properties. We hypothesized that HS deficient tendons should be less capable of undergoing effective repair when challenged by a traumatic insult (tenotomy) due to an inability of the mutant perlecan from participating in HS dependent interactions with growth factors such as FGF-2 to promote reparative cell proliferation and matrix synthesis as has been shown in impaired vascular wound healing *in Hspg2* exon 3 null mice (*Zhou et al., 2004*). *Hspg2* exon 3 null mice also lay down significantly lower levels of TGF-β in tissues thus this important anabolic growth factor is unavailable to participate in such tissue repair processes in this genotype (*Shu, Smith & Melrose, 2016*).

In the present study we examined murine tail and Achilles tendon from C57BL/6 and *Hspg2* exon 3 null mice employing biomechanical, biochemical and molecular methods and imaged tendons by immunolocalising type VI collagen and perlecan and collagen fibril organization by transmission electron microscopy (TEM).

## MATERIALS AND METHODS

Ethics approval for this study was obtained from The Animal Care and Ethics Review Board of The Royal North Shore Hospital, St. Leonards, Sydney, Australia. (RNS/UTS 0709-035A J Melrose, C Little, R Appleyard. Evaluation of $\Delta3-/\Delta3-$ *HSPG2* HS deficient mice).

### Tissues

$Hspg2^{\Delta3-/\Delta3-}$ homozygous mouse breeding pairs backcrossed into a C57BL/6 background for 12 generations were kindly supplied by Dr. R Soinninen, University of Oulu BioCentre, (Oulu, Finland). WT C57BL/6 mice were obtained from Jackson Laboratories (Bar Harbor, ME, USA). All mice were caged in groups ($n = 2$–5 mice per 500 cm$^2$ cage floor space) and received acidified water and complete pelleted food *ad libitum*. All cages were individually ventilated with filter lids, sterilised Aspen chip bedding, environmental enrichment (tissues, house) and maintained at 20–22 °C, 50–60% humidity, with a 12 h light-dark cycle regimen. Only male mice were used for these studies.

### Overview of the analyses undertaken in this study

1. Genotyping of WT and Perlecan exon 3 null mice.
2. Weight of WT and Perlecan exon 3 null mice up to 20 weeks of age.
3. HS contents of purified perlecan samples measured by ELISA.
4. Material properties (UTS/TM) of tail and Achilles tendons over 3–12 weeks of age.
5. Compositional analyses of the GAG and hydroxyproline contents of tail tendons in WT and Perlecan exon 3 null mice up to 12 weeks of age.
6. Material properties of control non operated Achilles tendons over 2–8 weeks of age and the recovery of material properties (UTS/TM) of tenotomised Achilles tendons in WT and Perlecan exon 3 null mice over 2–8 weeks post operatively.
7. Collagen I, versican, biglycan, lumican, decorin, perlecan, LTBP-1 and 2, fibrillin-1, elastin gene expression by 3–12 week old WT and Perlecan exon 3 null tail tenocytes.
8. Response of WT and Perlecan exon 3 null mouse tail tenocyte monolayer cultures to FGF-2 (1–100 ng/ml): *Mmp2, Mmp3, Mmp13, Adamts4, Timp2, Timp3* gene expression.
9. Transmission electron microscopy of collagen fibril cross-sectional areas in tail and Achilles tendon in WT and Perlecan exon 3 null mice at 3 and 12 weeks of age.

### Genotyping of $Hspg2^{\Delta3-/\Delta3-}$ mice

Genomic DNA was extracted from WT and $Hspg2^{\Delta3-/\Delta3-}$ mouse tail tips using commercial kits (Qiagen). Specific regions of the mouse perlecan gene were amplified by PCR using genotyping primers recognising intron 2 of mouse *Hspg2* (GTA GGG ACA CTT GTC ATC CT), exon 3 (CTG CCA AGG CCA TCT GCA AG) and $Hspg2^{\Delta3-/\Delta3-}$ (AGG AGT

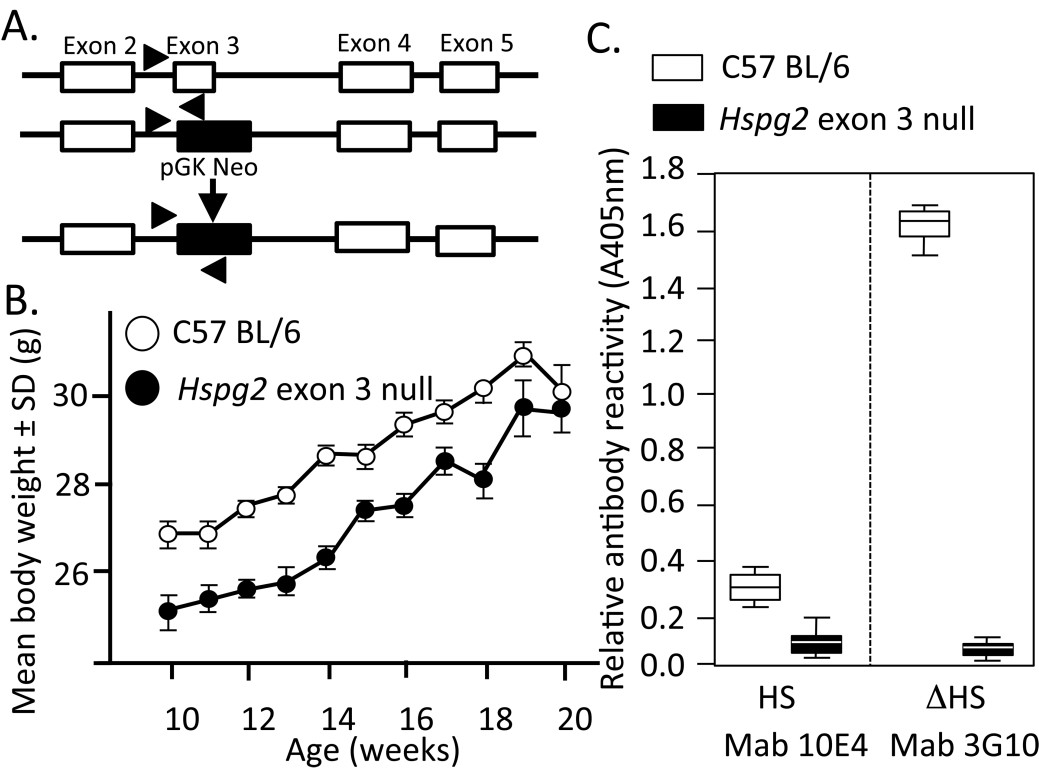

**Figure 1** **Genomic organisation, weights of perlecan exon 3 null and wild type mice and HS contents of perlecan from the two mouse genotypes.** Genomic organisation of exons 2–5 of the WT and $Hspg2^{\Delta 3-/\Delta 3-}$ mice (A). Body weights of male WT (open symbols) and $Hspg2^{\Delta 3-/\Delta 3-}$ mice (closed symbols) from 10–20 weeks of age (B). ELISA analysis of perlecan GAG side chains using MAb 10E4 to native HS and MAb 3G10 to the $\Delta$ HS stub epitope generated by heparitinase III pre-digestion demonstrating an absence of HS in the mutant perlecan (C).

AGA AGG TGG CGC GAA GG). The PCR products were identified by electrophoretic separation on 2% w/v agarose gels (Fig. S1A).

## Weights of WT and Perlecan exon 3 null mice

Mouse weights were measured in each genotype over 10–20 weeks of age (Fig. 1B).

## Isolation and identification of perlecan from skeletal muscle

Muscle from the hind limbs of two WT and two *Hspg2* exon 3 null mice were finely minced and extracted with 6M urea 50 mM Tris–HCl pH 7.2 (15 ml/g tissue) for 48 h at 4 °C. Perlecan was isolated using Resource Q anion exchange FPLC and electrophoresed on pre-poured 3–8% PAG Tris-acetate gradient gels, blotted to nitrocellulose and perlecan identified using MAb H300 (Santa Cruz Biotechnology, Dallas, TX, USA) (Fig. S1B). The GAG side chains of these samples were analysed by ELISA using MAb 10-E-4 and 3-G-10. Selected samples were pre-digested with Heparitinase III to generate $\Delta$-HS stub epitopes reactive with MAb 3-G-10.

## Biomechanical assessment of murine tail and Achilles tendons

Tail tendons from 3, 6 and 12 week old mice, ($n = 6$–8 at each age point), were dissected from underlying bone and connective tissue within 1 h of death. Mouse tails contain two dorsal and two ventral tendons. Ventral tendons were used exclusively in this study. These were dissected out approximately 5 mm from the base of the tail. Segments of 20–30 mm were used for biomechanical testing. A single ventral tendon was used for biomechanical testing per animal. The cross-sectional area of the tendon was measured using a custom micrometer device (*Choi et al., 2016*). Achilles tendons attached to the calcaneous and gastrocnemius muscles proximally were also collected. Biomechanical testing was undertaken in an Instron 8874 servo-hydraulic material testing apparatus. Tail tendons were marked with two reference points using Alcian blue and anchored in custom-built brass clamps. The distance between the clamps was 10 mm. With Achilles tendons the calcaneous was anchored in the lower clamp, and gastrocnemius muscle attached to the upper clamp. Tendons were loaded to failure at a rate of 1 mm/second (10% strain), real-time videos of each test were recorded at 100 frames per second using a high-speed camera mounted perpendicular to the tendon (Marlin F/145B camera; Allied Vision Technology, Newburyport, MA, USA). The videos of tendon deformation were analysed using LabView software (National Instruments, Austin, TX, USA) and the data normalised to tendon cross-sectional area to calculate stress. Elastic modulus was calculated from the gradient of the linear region on the stress–strain curve.

## Tendon compositional analyses

Finely-diced tail tendons were papain digested and sulfated GAG determined using 1,9-dimethylmethylene blue, bovine tracheal CS was used as standard (*Farndale, Buttle & Barrett, 1986*). Aliquots of the tissues were also hydrolysed in 6M HCl for hydroxyproline determinations by the dimethylaminobenzaldehyde procedure (*Stegemann & Stalder, 1967*).

## Achilles tendon tenotomy

Under general anaesthesia (2% isofluorane) Achilles tendons of 12 week old mice from each genotype were sharply transected mid-way between the calcaneal attachment and muscle leaving the plantaris tendon intact. The skin incision was closed with a subcutaneous Vicryl 8/0 suture and sealed with cyanoacrylate tissue glue. Mice were returned to their pre-operative social groups post-tenotomy. Groups of mice ($n = 10$ for each time point) were sacrificed at 2, 4 or 8 weeks post operatively (PO) and Achilles material properties measured.

## Gene expression in tail tendons

Mouse tail tendons from 3, 6, and 12 week old WT and $Hspg2^{\Delta 3-/\Delta 3-}$ mice were pooled ($n = 6$ at each age point) to provide ~50 mg wet weight of tissue for each RNA isolation. Tendons were snap frozen in liquid nitrogen, freeze-shattered using a Mikro dismembranator (B. Braun Biotech International, Melsungen, Germany) and total RNA extracted using Trizol (Invitrogen, Mulgrave, VIC, Australia), purified using Qiagen RNeasy columns (Qiagen, Chadstone Centre, VIC, Australia) and quantified by

NanoDrop (ThermoFisher Scientific, Scoresby, VIC, Australia). RNA (1 µg) from each sample was reverse transcribed using Omniscript Reverse Transcription Kit (Qiagen) with random pentadecamers (50 ng/ml; Sigma-Genosys, Castle Hill, NSW, Australia) and RNase inhibitor (10 U per reaction, Bioline, Sydney, NSW, Australia). The cDNA was subjected to qRT-PCR in a Rotorgene 6000 (Qiagen) using Immomix (Bioline, Sydney, NSW, Australia), SYBR Green I (Cambrex Bioscience, Rockland, ME, USA), and 0.3 µM validated murine-specific primers. Relative copy numbers for genes of interest were determined using a standard curve generated from pooled cDNA normalised to *Gapdh*. PCR primer specificity was confirmed by sequencing (SUPAMAC, Sydney University). Genes, primers, and annealing temperatures are listed in Table 1.

## Tenocyte monolayer cultures stimulated with FGF-2

Three week old tail tendons from six mice of each genotype were macerated and cultured in 2 ml DMEM/10% FBS/2 mM L-glutamine under an atmosphere of 5% $CO_2$, with media changes every 3-4 days. After 2 weeks the tissue was removed and the attached cells detached with trypsin-EDTA and sub-cultured in fresh media. Passage 3 cells were cryopreserved ($10^7$ cells/ml, 0.5 ml aliquots) in 10% v/v DMSO, 20% v/v FBS in DMEM. Tenocytes were re-seeded in 6-well plates at $2 \times 10^5$ cells per well for 48 h. The cultures were washed three times in serum-free DMEM, and incubated in DMEM/1% v/v FBS containing 0, 1, 10 or 100 ng/ml FGF-2 (PeproTech Inc, Rocky Hill, NJ, USA) for 24 h. Total RNA was extracted using Trizol, 1 µg RNA was reverse-transcribed then qRT-PCR undertaken for *Mmp2, Mmp3, Mmp13, Timp1, Timp3 and Adamts4*.

## Transmission electron microscopy of tail and Achilles tendon

Three tendons from 3- and 12-week old WT and *Hspg2* $^{\Delta 3-/\Delta 3-}$ mice were washed in 0.1 M sodium cacodylate buffer containing 3 mM $CaCl_2$, 100 mM sucrose pH 7.4 and fixed in 2.5% v/v glutaraldehyde/0.5% v/v paraformaldehyde for 30 min at room temperature, 4 °C for 24 h, followed by storage in 70% v/v ethanol. The fixed tissues were trimmed and post-fixed in 2% w/v $OsO_4$ in 0.1M cacodylate buffer for 1–2 h at 4 °C followed by dehydration in serial graded ethanol washes (25%, 50%, 75%, 95%, 100%, 100%, all v/v). The tissues were infiltrated with Spurrs resin/ethanol (1:1) overnight then with two overnight infiltrations of 100% resin and polymerised at 60 °C for 48 h. Ultra-thin transverse tissue sections (70 nm) were cut using an Ultracut T microtome and transferred to copper grids (200 mesh). The specimens were stained/contrasted for 10 min with 2 % w/v uranyl acetate and Reynold's lead citrate (1.33 g lead nitrate, 1.76 g sodium citrate dihydrate, 5 ml 1 M NaOH, in 50 ml $H_2O$ final total volume). The specimens were examined in a JEOL1400 transmission electron microscope at 120 kV at 25,000 × magnification. The images were analysed using ImageJ (public domain Java-based image processing software developed by NIH) to determine fibril diameters. Three separate regions of each specimen were photographed and at least 400 fibrils were measured in each image. When the fibril had a non-circular configuration the diameter across the minimum axis was measured. The frequency distribution of the collagen fibril diameters was calculated as a percentage of the total fibril numbers measured.

**Table 1  Murine-specific primers.** Murine-specific primers designed using MacVector for real time PCR.

| Molecule (gene) | Mouse accession # | Sequence 5′ to 3′ | Annealing temp (°C) | Product size (bp) |
|---|---|---|---|---|
| **Extracellular matrix proteins** | | | | |
| Collagen I (*Col1a1*) | X06753 | F TCT CCA CTC TTC TAG TTC CT<br>R TTG GGT CAT TTC CAC ATG C | 55 | 269 |
| Versican (*Vcan*) | XM994074 | F ATG ATG GGG AAG GAA GGG GTT C<br>R AGC CAG CCG TAA TCG CAT TG | 57 | 236 |
| Biglycan (*Bgn*) | L20276 | F ACT TCT GTC CTA TGG GCT TCG G<br>R GCT TCT TCA TCT GGC TAT GTT CCT C | 57 | 218 |
| Lumican (*Lum*) | NM008524 | F TAC AAC AAC CTG ACC GAG TCC G<br>R CGA GAC AGC ATC CTC TTT GAG C | 55 | 159 |
| Decorin (*Dcn*) | NM007833 | F CAA CAA CAA ACT CCT CAG GGT GC<br>R TTG CCG TAA AGA CTC ACA GCC G | 57 | 165 |
| **Elastin and associated proteins** | | | | |
| Perlecan (*Hspg2*) | NM008305 | F TCT GTC TGC CTG GCT TCT CT<br>R CGA ATT CAA TTG TCT CGG GT | 56 | 204 |
| LTBP-1 (*Ltbp1*) | NM019919 | F GGG AGC ATC TGA GTG AGG AG<br>R TCA CAG GGA TAT TGC ACA GC | 56 | 165 |
| LTBP-2 (*Ltbp2*) | NM013589 | F CAC CCA GAC CAG CCT TCC CA<br>R AGT CCT TGC AGA GGC CCA GG | 57 | 126 |
| Fibrillin-1 (*Fbn1*) | NM007993 | F ATC CGC TGT ATG AAT GGG GG<br>R CTG GCA CAT CTG GTT GCT TAC C | 58 | 228 |
| Elastin (*Eln*) | NM007925 | F AGC CAA ATA TGG TGC TGC TG<br>R GGG TCC CCA GAA GAT CAC TT | 58 | 246 |
| **MMPs/TIMPs** | | | | |
| MMP-2 (*Mmp2*) | NM008610 | F ATT TGG CGG ACA GTG ACA CCA C<br>R ATC TAC TTG CTG GAC ATC AGG GGG | 59 | 231 |
| MMP-3 (*Mmp3*) | NM010809 | F GCT GAG GAC TTT CCA GGT GTT G<br>R GGT CAC TTT TTT GGC ATT TGG GTC | 53 | 120 |
| MMP-13 (*Mmp13*) | NM008607 | F GAT GAC CTG TCT GAG GAA G<br>R ATC AGA CCA GAC CTT GAA G | 55 | 357 |
| ADAMTS-4 (*Adamts4*) | NM172845 | F TAA CTT GAA TGG GCA GGG GGG TTC<br>R AAT GGC TTG AGT CAG GAC CGA AGG | 60 | 245 |
| TIMP-1 (*Timp1*) | BC051260 | F ATC TCT GGC ATC TGG CAT CCT C<br>R GGT GGT CTC GTT GAT TTC TGG G | 56 | 154 |
| TIMP-3 (*Timp3*) | NM011595 | F ATT ACC GCT ACC ACC TGG TTT G<br>R TCT GGG AAG AGT TAG TGT CTG GGA C | 58 | 198 |
| GAPDH (*Gapdh*) | BC083149 | F TGC GAC TTC AAC AGC AAC TC<br>R CCT GCT CAG TGT CCT TGC TG | 55 | 200 |

## Statistical analyses

Groups of paired parametric data from mechanical measurements, hydroxyproline and sGAG compositional data and fibril diameters measured by TEM were compared by unpaired Students-t test for differences between age and genotypes. Non-Gaussian data from qRT-PCR analyses were analysed using Kruskal-Wallis then Mann–Whitney U ranked tests. The alpha level was set at 0.05.

## RESULTS

### Genotyping of mouse strains

Figure 1A depicts the replacement of perlecan exon-3 with a pGK-neo cassette in the $Hspg2^{\Delta3-/\Delta3-}$ mice. $Hspg2^{\Delta3-/\Delta3-}$ mice were fertile and litters were of expected size. They had no gross abnormalities or difference in appearance compared to WT mice at birth. By 3 weeks of age, the previously reported micropthalia in $Hspg2^{\Delta3-/\Delta3-}$ animals was evident. Age-matched $Hspg2^{\Delta3-/\Delta3-}$ mouse body weights were less than corresponding WT mice 10 to 18 weeks of age (Fig. 1B). Although smaller, $Hspg2^{\Delta3-/\Delta3-}$ mice had similar skeletal proportions to WT mice and no apparent musculoskeletal abnormalities. Mutant mice were more docile when handled but no other behavioural abnormalities were noted other than small leaky eyes.

### Perlecan isolated from C57BL/6 Wild Type and Hspg2 exon 3 null mice

Perlecan isolated from skeletal muscle demonstrated a core protein of ~420 kDa for WT perlecan while mutant perlecan had a molecular weight of ~300–390 kDa (Fig. S1B). ELISA analysis demonstrated that perlecan from C57BL/6 Wild Type mice contained HS chains while $Hspg2$ exon 3 null perlecan did not (Fig. 1C).

### Tendon material properties and biochemical composition

Ultimate tensile stress (UTS) (Figs. 2A, 2C) and tensile modulus (TM) (Figs. 2B, 2D) measurements of tail (Figs. 2A, 2B) and Achilles (Figs. 2C, 2D) 3–12 week old tendons demonstrated there were no significant difference in 3-week old tendons. UTS and TM values for tail and Achilles tendons increased with maturation (Figs. 2A–2D). Six to twelve week old tail tendons from the perlecan exon 3 null mice displayed moderately greater UTS and TM values compared to WT tail tendons ($P < 0.05$) (Figs. 2A, 2B). Tail tendon sGAG levels underwent a maturation-dependent decline in $Hspg2^{\Delta3-/\Delta3-}$ mice (Figs. 2E, 2F), with 50% reduction in GAG content at 12 weeks compared to 3 weeks (3 >6 >12 week), and lower GAG content in $Hspg2^{\Delta3-/\Delta3-}$ mice whereas hydroxyproline contents did not change significantly with age or genotype.

### Achilles tendon tenotomy

With tissue maturation the non-operated contralateral perlecan exon 3 null Achilles tendons displayed moderately higher UTS and TM values by 20 weeks of age compared to the WT tendons ($P < 0.05$). Following tenotomy, WT and perlecan exon 3 null Achilles tendons partially recovered their UTS and TM material properties over an 8 week recovery period reaching approximately 50% of the UTS/TM values of the non-operated contralateral tendons at 20 weeks of age (Figs. 3A, 3C). There was a moderately increased trend in the rate of recovery of UTS/TM in the WT mice compared to the perlecan exon 3 null mice following tenotomy. The UTS and TM values for the WT tendons however were more variable than the perlecan exon 3 null tendons. Statistics on the mean data at each time point for each genotype indicated this trend was only moderately elevated in the WT Achilles tendons with a $P < 0.05$ (Figs. 3B, 3D).

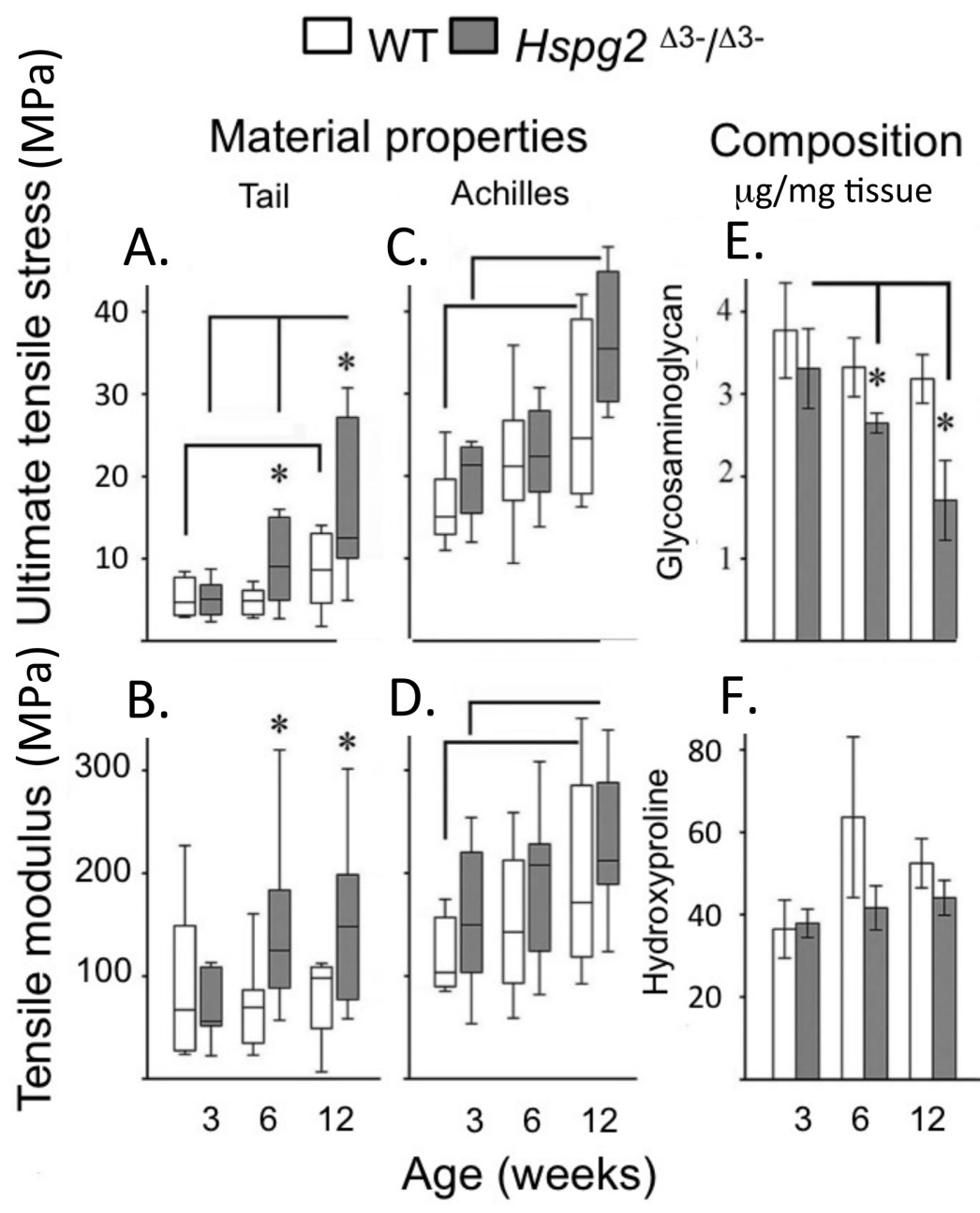

**Figure 2  Material properties of wild type and perlecan exon 3 null mouse tail and Achiles tendons and compositions.** Tendon material properties: ultimate tensile stress (A, C) and tensile modulus (B, D). GAG (E) and Hydroxyproline (F) content of 3 to 12 week-old WT and $Hspg2^{\Delta3-/\Delta3-}$ tail tendons. Box plots show mean (line in box) and data range (box 25–75%, whiskers maximum–minimum). White bars: WT; gray bars: $Hspg2^{\Delta3-/\Delta3-}$. Bar graphs show mean ± standard error of mean. Brackets–$P < 0.05$ between samples, * $P < 0.05$ between genotypes. $N = 6$–8 for each sample.

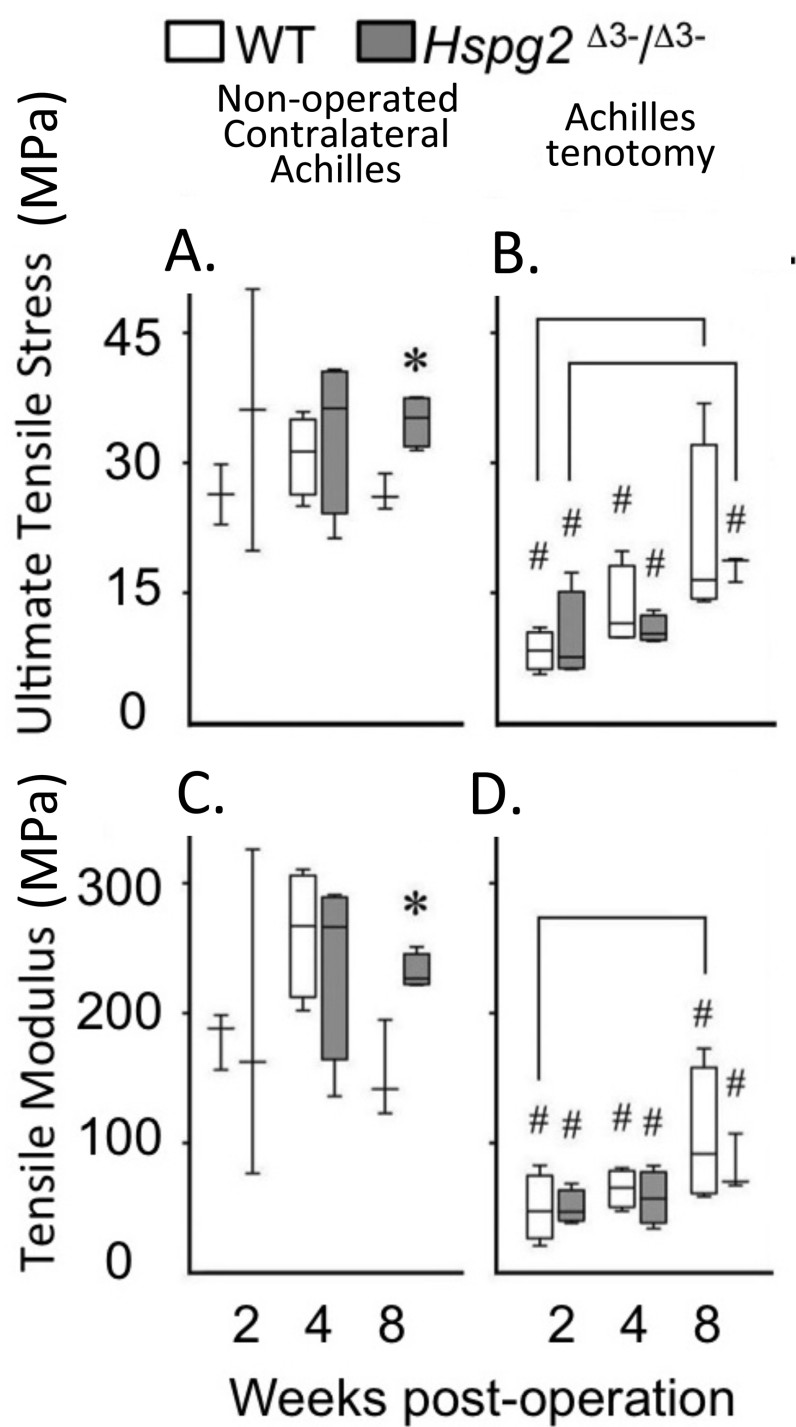

**Figure 3 Material properties of Achilles tendons following tenotomy and recovery for up to 8 weeks post surgery.** Achilles tendon material properties at 2, 4 and 8 weeks after surgical tenotomy: normal contralateral (A, C) and surgical tenotomy (B, D), equivalent to 14, 16 and 20 weeks of age. White bars: WT; gray bars: $Hspg2^{\Delta3-/\Delta3-}$. Box plots show median (line in box), inter-quartile range (box) and data range (whiskers, maximum–minimum). Bracket–$P < 0.05$ between samples. * $P < 0.05$ between genotypes. # $P < 0.05$ to contralateral. $N = 6$–8.

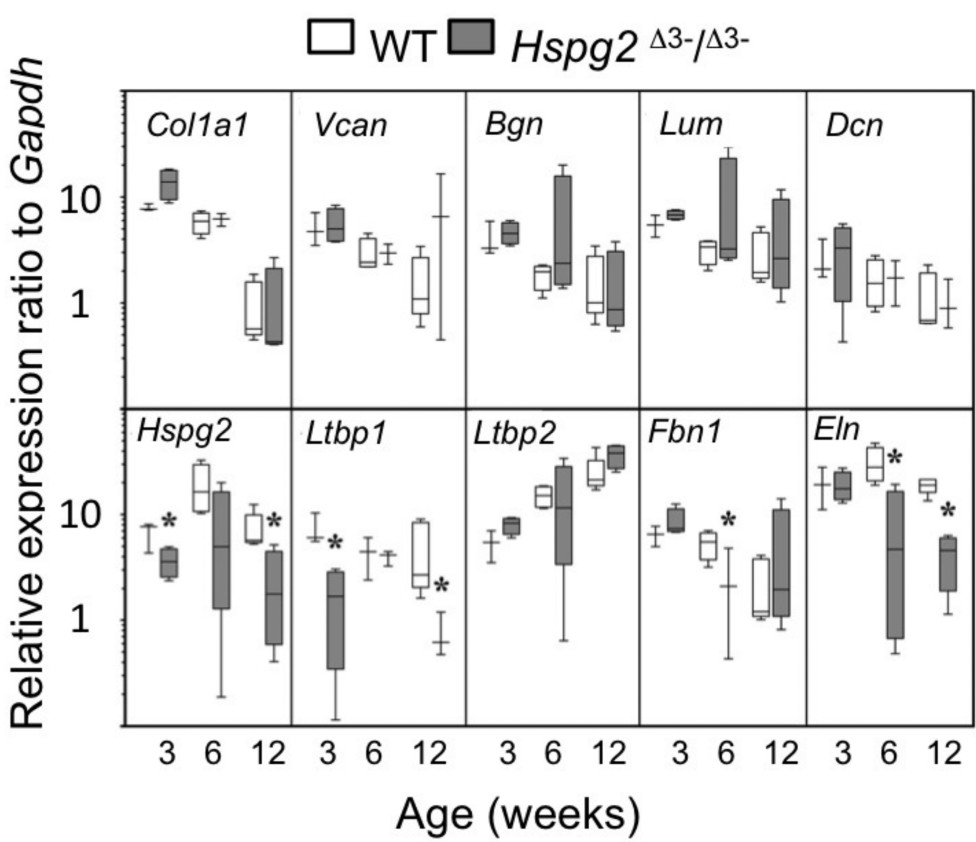

**Figure 4** **Comparative gene expression in wild type and perlecan exon 3 mouse tail tendon.** Comparative gene expression of selected extracellular matrix genes and elastin-associated protein genes in mouse *ex vivo* tail tendons at 3, 6 and 12 weeks old. * $P < 0.05$ between genotype. Data were normalised to *Gapdh* expression. White bars: WT; gray bars: $Hspg2^{\Delta3-/\Delta3-}$. Box plots show mean (line in box), interquartile range (box) and data range (whiskers, maximum–minimum). $N = 6$ per sample.

## Tail tendon gene expression

qRT-PCR of selected ECM genes in mouse tail tendons, (*Col1a1*, *Vcn*, *Bgn*, *Dcn*), demonstrated a maturation-dependent decline in gene expression over 3 to 12 weeks in both genotypes (Fig. 4). Relative gene expression levels for perlecan core protein (*Hspg2*) and the elastin micro-fibril associated proteins *Ltbp1*, *Fbn1*, and *Eln* were significantly lower in $Hspg2^{\Delta3-/\Delta3-}$ mice compared to age-matched WT mice. *Ltbp2* gene expression displayed an increase with ageing in both genotypes.

## Tendon outgrowth cell responses to FGF-2

*Mmp2, Mmp3, Mmp13* and *Adamts4* expression were significantly higher in basal $Hspg2^{\Delta3-/\Delta3-}$ mice compared to WT tenocyte cultures (0 ng/ml FGF-2). *Mmp2* expression in $Hspg2^{\Delta3-/\Delta3-}$ cultures remained significantly higher than WT at all doses of FGF-2 examined. *Mmp* 3 and *Mmp13* expression increased dose-dependently in both genotypes, and this response was greater in $Hspg2^{\Delta3-/\Delta3-}$ tenocytes at high doses of FGF-2 (*Mmp* 3 33–36 fold versus 50 fold and *Mmp13* 134–192 fold versus 226–248 fold at 10 and 100 ng/ml

in WT and $Hspg2^{\Delta3-/\Delta3}$) (Fig. 5). *Adamts4* gene expression was significantly decreased by increasing concentrations of FGF-2 treatment in $Hspg2^{\Delta3-/\Delta3-}$ tenocyte cultures but still remained significantly greater than WT cultures at all doses. FGF-2 up-regulated *Timp1* gene expression, less so in $Hspg2^{\Delta3-/\Delta3-}$ compared to WT cultures, favoring a pro-catabolic phenotype in the mutant mice. Expression of *Timp3*, the naturally occurring inhibitor of the ADAMTS (**A D**isintigrin **A**nd **M**etalloprotease with Thrombospondin motifs) enzymes, while equivalent in basal culture, was decreased to a greater extent by FGF-2 in $Hspg2^{\Delta3-/\Delta3-}$ tenocyte cultures at the highest dose.

## Measurement of Tail and Achilles tendon collagen fibril diameters by transmission electron microscopy

Figure 6 shows the changes in collagen fibril diameter of tail and Achilles tendon measured from TEM images of 3 and 12 week old mouse tendons. In 3 week old mice there were small differences between genotypes in mean fibril diameter ($\pm$SD) in tail (WT $= 202 \pm 60$ nm, $Hspg2^{\Delta3-/\Delta3-} = 193 \pm 46$ nm; not significant) and Achilles (WT $= 160 \pm 44$ nm, $Hspg2^{\Delta3-/\Delta3-} = 139 \pm 37$ nm; $P < 0.001$) tendons (Figs. 6A, 6B). This was reflected in minor differences in frequency distributions of collagen fibril diameters in the two genotypes in these immature mice (Figs. 6C, 6D). An increase in average collagen fibril diameter was evident from 3 to 12 week old in the WT tail ($202 \pm 60$ nm to $243 \pm 81$ nm; $P < 0.0001$) and Achilles ($160 \pm 44$ nm to $210 \pm 63$ nm; $P < 0.001$) tendons, accompanied by an increase in collagen fibril diameter distribution in both tendons (Figs. 6C, 6D). Mean Achilles collagen fibril diameter also increased from 3–12 week old in $Hspg2^{\Delta3-/\Delta3-}$ mice ($139 \pm 37$ nm to $150 \pm 34$ nm $p < 0.001$) to a lesser extent than in WT mice (8 versus 31%), and significantly decreased in $Hspg2^{\Delta3-/\Delta3-}$ tail tendons ($193 \pm 46$ nm to $84 \pm 28$ nm; $P < 0.001$). Thus differences between genotypes in mean collagen fibril diameter and distribution in tail and Achilles tendons were more marked by 12 weeks of age.

## DISCUSSION

This study uncovered some surprising findings regarding the potential role of perlecan HS chains in the functional properties of tail and Achilles tendon. We hypothesised that ablation of the HS chains of perlecan should have a significant effect on ECM organization and would also be reflected in tendon material properties. The HS chains of perlecan also interact with growth factors and present these to their cognate receptors to modulate cell signaling, proliferation, differentiation and matrix synthesis (*Whitelock, Melrose & Iozzo, 2008*). Thus we expected that ablation of perlecan HS chains would also disrupt normal tissue homeostasis. None of these possibilities actually happened. Perlecan was selected as a target proteoglycan to investigate in this study since it is a major, ubiquitous, HS-proteoglycan with established functional attributes in connective tissues (*Whitelock & Melrose, 2011*). Other extracellular HS substituted proteoglycans such as agrin and type XVIII collagen and the cell associated glypican and syndecan families have also been identified (*Iozzo, 1998*; *Iozzo & Schaefer, 2015*). Type XVIII collagen and the glypican family of proteoglycans have not been identified in the main substance of tendon. Agrin

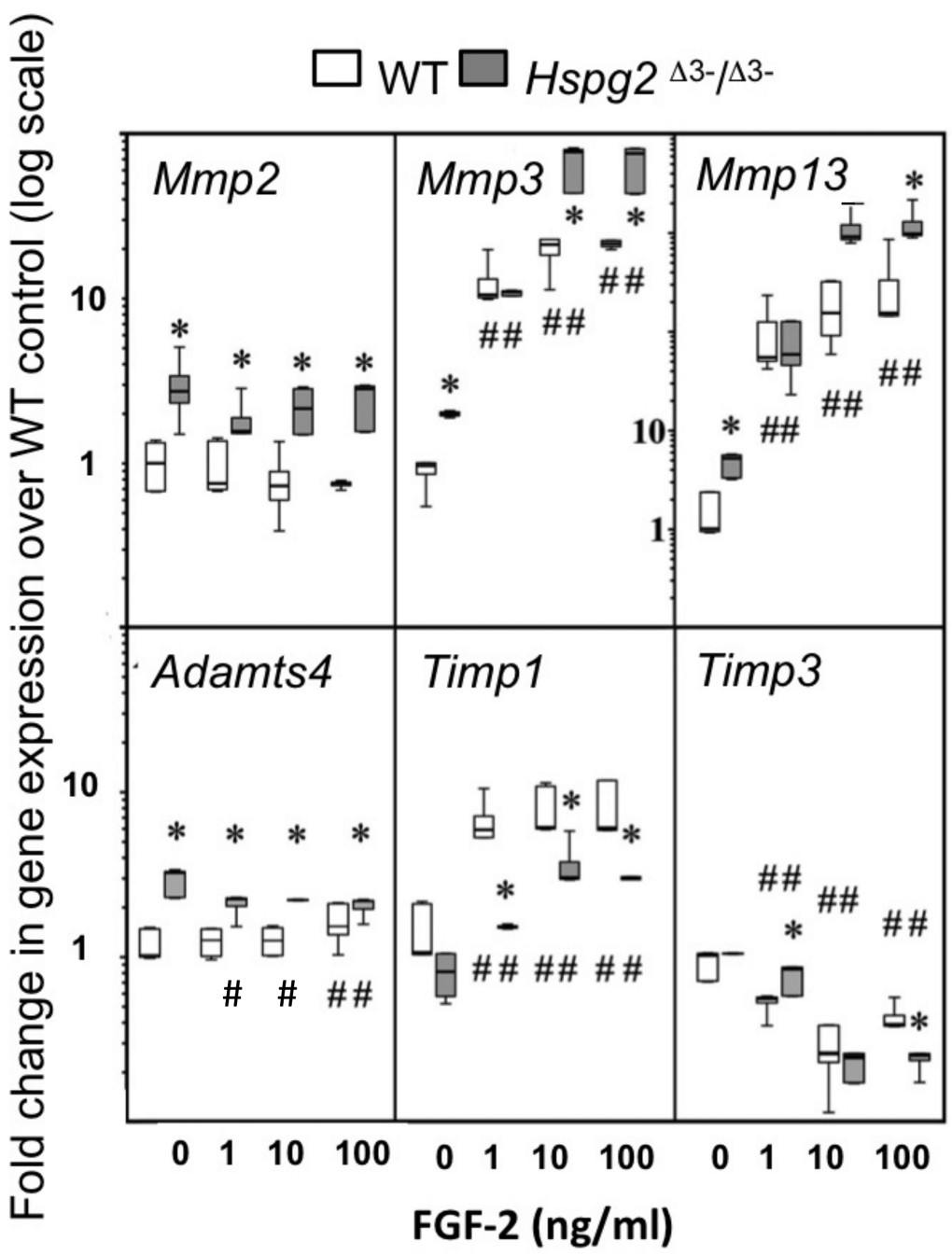

**Figure 5 Influence of FGF-2 on cultured wild type and perlecan exon 3 null tail tenocytes in mono-layer culture.** Gene expression in 3 week-old mouse tail tendon outgrowth tenocytes cultured with FGF-2 (0, 1, 10 or 100 ng/ml). Data expressed as a fold change relative to the expression of the WT control (0 ng/ml FGF-2). White bars: WT; gray bars: $Hspg2^{\Delta 3-/\Delta 3-}$. Box plots show mean (line in box), interquartile range (box) and maximum–minimum (whiskers). $N = 6$ for each sample. # $P < 0.05$ compared to untreated control; ⋆ $P < 0.05$ between genotypes at that concentration of FGF-2. Note $Mmp13$ $Y$-axis differs from all other genes with range from 0.1–10,000× instead of 0.1–100×.

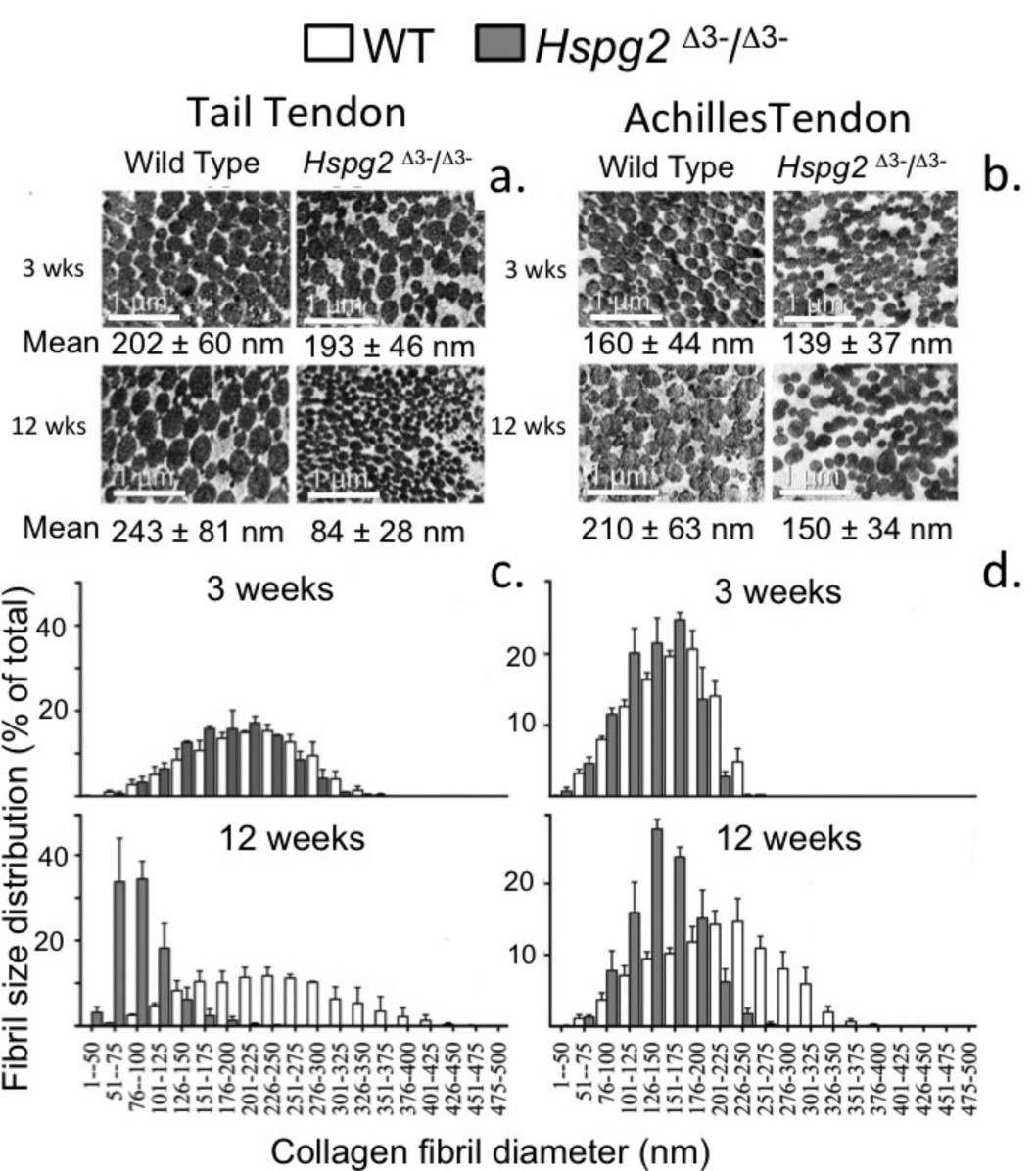

**Figure 6 Measurement of the collagen fibril diameters in wild type and perlecan exon 3 null tail and Achilles tendons by transmission electron microscopy.** Measurement of the collagen fibril diameters in cross-section of 3 and 12 week-old mouse tail and Achilles tendons from transmission electron microscopy (TEM) images. Representative TEM images (A, B) are shown with respective mean ± standard deviations. Fibril diameter data were categorised into defined nanometer ranges (x axis) giving the overall size distribution (C, D). The frequencies were expressed as a percentage of the total fibril numbers counted (y axis). Three tendon samples from three animals per age/genotype were measured. Bar graph shows mean ± standard deviation. White bars: WT; gray bars: $Hspg2^{\Delta3-/\Delta3-}$.

is an extracellular HS-proteoglycan of the muscle–tendon neuromuscular junction (NMJ) and is also found in articular cartilage (*Finni, 2006*). FGF-2-agrin interactions modulate neurite outgrowth in the NMJ and stabilize axonal development (*Cotman, Halfter & Cole, 1999*; *Kim et al., 2003*). Syndecan was shown to be produced by tenocytes in a proteomic study (*Sato et al., 2016*; *Smith, Thomas & Birk, 2012*), by fibroblasts (*Sawaguchi et al., 2006*) and chondrocytes (*Melrose et al., 2012*) and participates in FGF-2 mediated cell signalling (*Salmivirta & Jalkanen, 1995*). Members of the syndecan family and agrin are therefore candidate HS-proteoglycans which could potentially act in a "fill-in" capacity in HS deficient perlecan exon 3 null tenocytes. NG2/CSPG4 is another transmembrane proteoglycan produced by cells of a chondrogenic background such as tenocytes. CSPG4 binds FGF2 resulting in cell proliferation (*Cattaruzza et al., 2013*; *Takeuchi et al., 2016*); it also acts as a type VI collagen receptor and binds to perlecan (*Tang et al., 2018*). Type VI collagen/CSPG4 regulate tenocyte behavior, play important roles in the assembly of tendon matrix, and have crucial roles in tendon repair processes (*Sardone et al., 2016*). CSPG4 is therefore another candidate proteoglycan worthy of consideration in the FGF-2 mediated responses we observed in the present study. However, it is not known to what extent tenocyte CSPG4 undergoes cell signalling in response to FGF-2. Upon FGF binding to microvascular cells, the cytoplasmic domain of CSPG4 is phosphorylated, but there is no evidence that this event elicits signal transduction in a similar way to how this occurs in perlecan-HS-FGF-FGFR mediated interactions (*Sardone et al., 2016*). It therefore remains to be established to what extent CSPG participates in tenocyte responses in the present study. *Wnt*-signalling is a highly conserved and tightly controlled cascade of signal transduction pathways, essential for articular cartilage development and joint homeostasis (*Chun et al., 2008*). Syndecan-4 (Sdc4) is the most abundant syndecan family member in articular cartilage (*Echtermeyer et al., 2009*). Sdc4 and agrin facilitate *Wnt*-signalling in articular chondrocytes, but have opposing actions: Sdc4 promotes *Wnt* signaling, reduces the expression of *Acan* and *Col2A1* and inhibits GAG production (*Van den Berg, 2011*), agrin inhibits *Wnt* signalling and promotes chondrogenesis (*Eldridge et al., 2016*; *Hausser et al., 2007*). Syndecan-4 controls the activation of ADAMTS-4 and 5 either by direct interaction with these proteases or indirectly by regulation of MAPK dependent synthesis of MMP-3 (*Echtermeyer et al., 2009*). The initiating role of ADAMTS-5 in proteoglycan loss in OA appears dependent on interaction with the transmembrane proteoglycan Sdc-4. Sdc-4 deficient mice are less prone to cartilage changes induced by experimental OA and display reduced ADAMTS-5 activity (*Van den Berg, 2011*). Sdc4 also regulates collagen fibril architecture, condensation and orientation, proteoglycan production and remodelling. Tenocytes and fibroblasts both participate in *Wnt* signaling and express *Wnt3a*, β-*catenin*, *Lrp5* (Low-density lipoprotein receptor-related protein 5) and *Tcf1* (hepatocyte nuclear factor-1 alpha) (*Liu et al., 2014*; *Liu et al., 1995*; *Liu et al., 2010*). The *Wnt* signalling pathway plays a vital role in pathological calcification in animal models of tendinopathy and in pathological human tendon (*Liu et al., 2013*). *Wnt3a* increases ALP activity, calcium nodule formation and expression of osteogenic markers in tendon derived progenitor stem cells. In equine tendon, Wnt/β-catenin signalling promotes the differentiation of bone marrow stromal Mesenchymal stem cells (MSCs) into tenomodulin

expressing tendon cells (*Miyabara et al., 2014*). Wnt/β-catenin signalling in 6 week old rat Achilles tendon however suppressed expression of the tenogenic genes Scleraxis (*Scx)*, Mohawk homeobox (*Mkx)*, and Tenomodulin (*Tnmd)* suggesting that Wnt/β-catenin signalling repressed tenogenic gene expression in rats (*Kishimoto et al., 2017*). These two studies illustrate the need for a better understanding of the regulatory mechanisms which control tendon development and remodelling and the extracellular factors which control tenocyte gene expression. The observations of the present study which showed that ablation of perlecan HS chains resulted in a more compactly organized tendon with increased tensile properties represent a further aspect of the functional organization of tendon tissue and tenocyte regulation.

The reduction in size of collagen fibrils we observed in the *Hspg2* exon 3 null mouse tail tendons was also an unexpected finding of the present study. In other knock-out studies where proteins which regulate tendon collagen fibrillogenesis are deleted, the collagen fibrils generally display an increased size (*Ameye et al., 2002*; *Chen et al., 2003*; *Danielson et al., 1997*; *Ezura et al., 2000*; *Jepsen et al., 2002*). Collagen VI knockout mice are an exception and small collagen fibrils similar in size to those observed in the present study are also observed (*Alexopoulos et al., 2009*). Another aspect of the hypothesis we tested in this study was whether a deficiency of perlecan HS chains resulted in an impaired ability of the tendons to undergo intrinsic repair processes due to the lack of a contribution from HS dependent growth factors. Skin wounds display incomplete vascular repair in *Hspg2* exon 3 null mice (*Zhou et al., 2004*). Tenotimised Achilles tendons from the *Hspg2* perlecan exon 3 null and WT mice also displayed a similar mean recovery rate in UTS and TM compared to non-operated contralateral Achilles tendons. This response was more variable in the WT mice and the trend in the recovery of material properties in the Achilles tendons was slightly slower in the *Hspg2* exon 3 null mice compared to WT mice but this was not statistically significant. Achilles tendons in both of the mouse genotypes had recovered around 50% of their UTS and TM by 12 weeks post tenotomy. Thus HS is not a strict requirement for tendon repair and in this respect repair processes in tendon differ from those operative in the repair of skin wounds in perlecan exon 3 null mice.

Shoulder instability is a clinical problem occurring due to tendon laxity, recurrent subluxations or in severe cases complete shoulder dislocation (*Walz, Burge & Steinbach, 2015*). This is a painful condition, which increases in severity if left untreated. Treatment options include surgical intervention, physiotherapy with specific strengthening exercises for muscle groups to better hold the shoulder in proper position, cortisone injections and anti-inflammatory medications (*Bigliani et al., 1996*; *Moon et al., 2011*). The findings of the present study indicate that digestion of the HS component of shoulder tendons with heparanase may improve their tensile properties making them more supportive in cases of shoulder laxity. Future research should aim to explore this possibility further.

## CONCLUSIONS

Ablation of perlecan HS chains was not detrimental to tendon function or intrinsic repair and moderately improved their tensile properties. The reduced size of collagen fibrils, and

a more condensed fibril packing density in HS deficient perlecan exon 3 null mice and reduced expression of elastin undoubtably contributed to the observed change in tendon material properties. Perlecan exon 3 null tenocytes displayed a catabolic phenotype with FGF-2. This may explain why an age dependent decline in sGAG content was observed in HS-deficient tendons. Agrin and syndecan-4 in tendon are putative "fill-in" proteoglycans which may have acted in place of HS free perlecan. Syndecan-4 already has established roles in the regulation of cartilage degradation through activation of ADAMTS-5 in OA.

## ACKNOWLEDGEMENTS

Dr. Joanna Peterson developed the tendon materials testing procedures used in this study as part of her PhD studies at The Murray-Maxwell Biomechanics Laboratory within the Institute of Bone and Joint Research of The Kolling Institute of Medical Research. Ms. Susan Smith expertly conducted the tendon immunolocalisations reported in this study.

### Funding

This work was supported by NHMRC Project Grant 1004032. The funders had no role in study design, data collection and analysis, decision to publish, or preparation of the manuscript.

### Grant Disclosures

The following grant information was disclosed by the authors:
NHMRC Project Grant: 1004032.

### Competing Interests

The authors declare there are no competing interests.

### Author Contributions

- Cindy C. Shu performed the experiments, prepared figures and/or tables, authored or reviewed drafts of the paper, approved the final draft.
- Margaret M. Smith conceived and designed the experiments, analyzed the data, approved the final draft, biostatistics, advice on qRTPCR.
- Richard C. Appleyard conceived and designed the experiments, performed the experiments, contributed reagents/materials/analysis tools, approved the final draft.
- Christopher B. Little conceived and designed the experiments, analyzed the data, authored or reviewed drafts of the paper, approved the final draft.
- James Melrose conceived and designed the experiments, analyzed the data, contributed reagents/materials/analysis tools, prepared figures and/or tables, authored or reviewed drafts of the paper, approved the final draft.

### Animal Ethics

The following information was supplied relating to ethical approvals (i.e., approving body and any reference numbers):

The University of Sydney Animal Care and Ethics Research Committee approved all animal procedures conducted in this study (RNS/UTS 0709-035A J Melrose, C Little, R Appleyard. Evaluation of $\Delta 3 - /\Delta 3 -$ HSPG2 HS deficient mice).

## Data Availability

The raw data are provided in a Supplemental File.

## Supplemental Information

Supplemental information for this article can be found online at http://dx.doi.org/10.7717/peerj.5120#supplemental-information.

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
