# Peer review of "Achilles and tail tendons of perlecan exon 3 null heparan sulphate deficient mice display surprising improvement in tendon tensile properties and altered collagen fibril organisation compared to C57BL/6 wild type mice"

_PeerJ, doi:10.7717/peerj.5120_

## Round 0.1 · original submission · Major Revisions

The interpretation of the mechanical data and altered responsiveness to FGF-2 in the Hspg2-deficient animals require more careful interpretation and the statistical analyses improved. The conclusions of the manuscript then require revision.

The structure and writing style of the article could also be improved and further information on the methods provided.

Reviewer 1 ·

Basic reporting

Writing:
The abstract is not very clearly written. Also in the introduction are some errors

Abstract
Abstract overall not very clearly written and not well structured
Line 31: Please explain all abbreviations. Eg HSPG2 = Heparan Sulfate Proteoglycan 2
Line 35: tenocytes isolated from where?
line 36. If you have measured mRNA you should call it mRNA instead of gene expression
lin37: …Fbn1 protein or mRNA expression? Unclear
line 41. Changes in UTS/TM biomechanics? Do you mean a reduction in UTS? Please spiecify
line 44: UTS measured how long after tenotomy? Also and more importantly, your data do not support a slower recovery of UTS in the HS deficient mice at all!
Line 46: again, your data do not show impaired material properties in deficient mice at all! If anything, HS deficiency improves stiffness and UTS!

Introduction
Line 56: That is not correct ,collagen isn’t a PG
Line 60: you named VEGF twice, and again please explain abbreviations (eg. VEGF, PDGF, SHH, BMP….)
Line 72: That is not correct, tendons are extensible, and they are more like a tough rubber band than like a cable.
Line 73:That is not correct, the connection between bone and bone is called a ligament, not a tendon!
Line96: Explain abbreviation AFM
Line97: …” which may be cryoprotective”. And? I am not quite sure how to deal with this information. What/why does it matter that it is cryoprotective. Please explain.
Line 109: very sudden appearance of FGF-2 here. You should elaborate a bit more on why you think FGF-2 has an essential role in the tendon healing process



Article structure:
Line300-309: These are not your results, but a description of the tendon structure. Thus, this does not belong here
Line 314: First time you mention that you used the ACL as positive control. This should have been stated in the methods

Experimental design

The research question itself is well defined. However, it is not clear why you are focussing on tendon
Line 63: Why tendon? Did you make the mouse and the only tissue showing measurable changes was the tendon, or did you aim for the tendon to start with? And if the latter, why? What about other tissues?
Line 111-113: since TGF-ß has an important function in the tenocytes response to mechanical stimuli and in the healing process, couldn’t any changes in tendon biomechanics not be an indirect effect of the HS deficient mice, being attributed to the missing TGF-ß?

Regarding the Methods there is some information missing. Also, I would highly recommend a description of the study design to start with, briefly outlining all experiments that are done
Line 139: why was perlecan isolated from skeletal muscle?
Line152-159: What was the distance between the clamps? Why didn’t you use the actuator displacement instead of video footage to determine tendon deformation? How did you determine tendon cross-sectional area?
Line 190: n=?
Line 198: n=?
Line 210: it is indeed debatable if GAPDH is a good housekeeping gene for tendon tissue. In my opinion, there exist better ones. I would prefer 18S, topo-1 or EIF4A2.

Regarding the experimental design, I very much like that data on gene expression, mechanical properties and histology are combined. That is very well done. However, I doubt that the stress deprivation works and I do not understand why the experiment with the sheep annulus fibrosus monolayer cultures was performed.
Line 167-170: n=?. Apart from being stress deprived, tenocytes are also exposed to all sorts of other changes to their natural environment. E.g exposing them to 10%FBS means far more nutrients and growth factors than they are usually exposed to.

Line 179-188: I do not quite understand what the immunolocalisation of type XI collagen in sheep annulus fibrosus monolayer cultures has to do with the rest of the experiments with mouse tail and mouse Achilles tendons?
Line 291-298: I still do not understand how these results fit in with the rest of the shown data

Validity of the findings

Statistics:
I have serious concerns regarding your statistics and recommend that you consult a statistician. You cannot use a T-Test if you are comparing more than 2 groups (e.g. mechanical properties at 3,6 and 12 weeks  you should have used an ANOVA instead)
Also If you have a very small sample size such as n=3 your data cannot be normally distributed, thus you cannot use a parametric test (e.g. fibril diameter n=3)
Figure 3 b: I very much doubt that the # indicating a difference in ADAMTS4 compared to untreated control in the WT is correct


Results:
Line 289: I do not agree at all with the statement, that the Achilles tendon of the knockout mice did not recover to the same extent as the wild type. The UTS mean values in the HS deficient mice are higher. Also this statement is not supported by statistics. There is no significant difference between genotypes.
Line 320-322: not exactly, expression was lower at some time points, e.g. Fbn1 expression was reduced at 6 weeks only, and given that there is no “box” for the HS deficient sample at 6 weeks I assume this might be a result of a very small sample size (n=3?)

Discussion:
Line362: Your data do not show detrimentally affected tendon material properties. Modulus and UTS are higher in the knockout mice, which means if anything, tendon properties are better in the deficient mice.

Line 383: how/where do you see a destabilization of tendon properties over time? Your data do not show that!

Line 386: Again, there are no deleterious changes in material properties

Line389: again, poor repair response is not supported by your data

Additional comments

I very much like that data on gene expression, mechanical properties and histology are combined. The authors have performed a range of very good experiments, which deserve to be published. However, I do not agree with the interpretation of the mechanical data and the statistics.

Also structure and writing style of the article could be improved and there is some information missing in the description of the methods. However, I think this could be fixed easily.

Reviewer 2 ·

Basic reporting

1. The manuscript is clearly written overall, but I think the meaning of ‘stress deprivation’ should be more clearly explained. I suggest including a sentence such as ‘Tendons were cultured in vitro in the absence of the physiological mechanical strain experienced by tendons in vivo. This is referred to as strain deprivation’.
2. Figure panels could be improved by labelling at the top left of each panel rather than at the top right.

Experimental design

Further characterisation of these previously generated Hspg2-deficient mice is of interest and has the potential to increase our understanding of the physiological role of perlecan. As such, the research described is of interest to the field.

Data showing that tendons from these animals exhibit altered mechanical properties (Figure 1d-i) are interesting, but I think should be discussed in more detail – see General Comments (point 2) below.

Validity of the findings

The authors conclude that tendons from mice with a Hspg2-deficiency were (line 418) ‘less responsive to FGF stimulation in terms of synthesis of matrix components (collagen, GAG)’ and that they (line 390) ‘demonstrated a disturbance in the ability .. to signal through FGF-2. FGF-2 failed to elicit an anabolic response.’ I think the results require more careful interpretation:

1. Since Figure 1i and Figure 3a and show no reduction in hydroxyproline content or expression of Col1a1, I assume the reduced collagen synthesis the authors refer to is shown in Figure 2? Is this really showing that less type IV collagen is synthesised, or does it reflect disturbed morphology of the Hspg2-deficient tendon?

2. I see no data that Hspg2-deficient tendons are ‘less responsive to FGF stimulation in terms of synthesis of matrix components (.. GAG)’. They exhibit less GAG with age (Figure 1h), but this is a reduction in in vivo levels, and cannot be attributed to altered responsiveness to FGF-2.

3. The implication is that FGF-2 signalling is impaired in Hspg2-deficient tendons. But Figure 3b show a much more complex picture. Basal expression (no added FGF-2) of Mmp2 is higher in Hspg2-deficient tendons, but there is no difference between the genotypes in Mmp2 expression in response to exogenously added FGF-2. Basal expression of both Mmp3 and Mmp13 is increased in response to FGF-2 in WT animals and this response is increased further in the Hspg2-deficient tendons. This suggests that responsiveness to FGF-2 is increased in these cases, not that it is decreased.

Additional comments

1. Line 265: The molecular weights of perlecan isolated from wild-type and Hspg2-deficient animals is referred to but not shown – these data should be included.
2. Line 272 and Figure 1d-i: Tendons from the tails of Hspg2-deficient animals lost GAG with age (Fig 1h), but exhibited increased ultimate tensile stress (Fig 1d) and tensile modulus (Fig 1f) with age compared to those from wild-type animals. Does this mean that Hspg2 deficiency increased the mechanical strength of the tendons? It would be useful to discuss the biological implications of these data in the Discussion.

Minor comments:
1. Line 180: The abbreviation ‘AF’ should be defined.
2. Line 180, 306: Please correct ug/ml.
3. Line 187: The abbreviation ‘diln’ should be written in full.
4. Line 190: The abbreviation ‘wk’ should be written in full.
5. Line 256: Should ‘micropthalia’ read ‘microphthalia’?
Line 257: Please include a reference to describing previous observation of microphthalia in Hspg2-deficient mice.
6. Line 298: Spelling of ‘localisation’ requires correction.
7. Figure legend 1: The meaning of the open boxes is not clear – see after ‘male WT’, ‘Hspg2 (D3-/D3-) mice’, and ‘MAb 3G10 to the’.
8. Figure legend 2: Please correct text to specify that ‘Perlecan C-terminal interaction with the a2b1 integrin and the a1 chain of type XI collagen’ – Greek symbols currently show as open boxes.
9. Figure legend 1: With reference to panels j-m, should the text rather read ‘normal contralateral (j, l) and surgical tenotomy (k, m)’ i. e. does the text refer to incorrect panels?
10. Figure 4: Please specify on the Figure that panel C refers to tail tendon, and D to the Achilles tendon (if this is correct).

---

## Round 0.2 · Major Revisions

The authors have responded to some of the major points of the first critique but still need to spell out clearly and concisely the outcomes of their study. What are the implications/possible reasons for effects of HS removal from perlecan on tendon stiffness

Reviewer 1 ·

Basic reporting

ok

Experimental design

ok

Validity of the findings

ok

Additional comments

Most of my concerns have been sufficiently addressed. The remaining minor points should be quick and easy to fix.
• I am still missing a brief description of the study design at the beginning of the methods. This would make it easier to understand what you have done.
• Also, you still have not added the information on the number of animals, tails or tissues to the methods description. This is important information which needs to be there. n= X should be added for every single experiment to make it transparent on how many samples the experiment was performed. Eg. “Tail tendons from 3, 6 and 12 week old mice“…. how many tail tendon fascicles from how many mice?
• Regarding the description of the mechanical testing, information on the distance between the clamps needs to be added to the text, as this has an effect on your results (the size of the end effect affecting strain)
• The section you now start the discussion with (line 289- 312), describing the function of certain proteins, seems utterly misplaced here and would better fit into the introduction. I would recommend to start your discussion with the section starting with “This study uncovered some surprising findings regarding…. “
• This section contains several errors, please delete: “Ultimate tensile strength (UTS) is a measure of a tissues maximum ability to withstand forces which attempt to elongate it while the tensile modulus (TM) is the ratio of stress (force per unit area) along a loading axis. UTS and TM are both measures of the stiffness of a material.“

Reviewer 2 ·

Basic reporting

Experiments are clearly described overall. Conclusions and Discussion requires more clarity - see General Comments below.

Experimental design

Clearly described.

Validity of the findings

1. Fig 2C concludes that UTS of Achilles tendons is not significantly changed by deletion of exon 3 of Hspg2, while Fig 3A concludes that UTS of Achilles tendons is significantly increased in these mutant mice. Please explain the discrepancy.

2. Line 231 - “ No notable differences were evident between the two mouse genotypes” in mechanical strength after tenotomy - does this mean there is no significant difference between the genotypes? Fig 3B shows comparisons with contralateral and between samples, but significance between genotype is not indicated.

3. Fig 4E and F are out of focus and low dpi, and thus not of publication quality. I am not convinced that Fig 4 adds anything to the manuscript.

4. Line 257: To improve clarity, the sentence could be reworded as “Adamts4 gene expression was significantly decreased by increasing concentrations of FGF-2..”

5. Fig 8 does not add to the manuscript - this is a text book representation of collagen fibril organisation and does not summarise the results obtained.

Additional comments

1. The manuscript can be more concisely written throughout. In particular, the Discussion requires significant truncation - a 12 page discussion is completely unwarranted, especially since the majority of the text is not relevant to the data or its interpretation. I would keep lines 390-413, 430-449, 550-571, and 611-624.

2. Despite the long Discussion, the results are not discussed or interpreted clearly and so the conclusions of this study are murky - deletion of HS chains from perlecan had no effect on tendon repair. Why does this deletion make the tendons stiffer? What is the reader supposed to conclude from the changes in mRNA levels shown?

---

## Round 0.3 · accepted · Accept

The major points raised by the Reviewers have been satisfactorily addressed

#